# Ambient Air Quality and Emergency Hospital Admissions in Singapore: A Time-Series Analysis

**DOI:** 10.3390/ijerph192013336

**Published:** 2022-10-16

**Authors:** Andrew Fu Wah Ho, Zhongxun Hu, Ting Zhen Cheryl Woo, Kenneth Boon Kiat Tan, Jia Hao Lim, Maye Woo, Nan Liu, Geoffrey G. Morgan, Marcus Eng Hock Ong, Joel Aik

**Affiliations:** 1Department of Emergency Medicine, Singapore General Hospital, Singapore 168753, Singapore; 2Pre-Hospital and Emergency Research Centre, Duke-NUS Medical School, Singapore 169857, Singapore; 3Duke-NUS Medical School, Singapore 169857, Singapore; 4Ministry of Health Holdings, Singapore 099253, Singapore; 5Environmental Quality Monitoring Department, Environmental Monitoring and Modelling Division, National Environment Agency, Singapore 228231, Singapore; 6Sydney School of Public Health, University of Sydney, Sydney, NSW 2006, Australia; 7Health Services & Systems Research, Duke-NUS Medical School, Singapore 169857, Singapore; 8Environmental Epidemiology and Toxicology Division, National Environment Agency, Singapore 228231, Singapore

**Keywords:** environmental epidemiology, air pollution, haze, air quality, emergency department, admission

## Abstract

Air pollution exposure may increase the demand for emergency healthcare services, particularly in South-East Asia, where the burden of air-pollution-related health impacts is high. This article aims to investigate the association between air quality and emergency hospital admissions in Singapore. Quasi-Poisson regression was applied with a distributed lag non-linear model (DLNM) to assess the short-term associations between air quality variations and all-cause, emergency admissions from a major hospital in Singapore, between 2009 and 2017. Higher concentrations of SO_2_, PM_2.5_, PM_10_, NO_2_, and CO were positively associated with an increased risk of (i) all-cause, (ii) cardiovascular-related, and (iii) respiratory-related emergency admissions over 7 days. O_3_ concentration increases were associated with a non-linear decrease in emergency admissions. Females experienced a higher risk of emergency admissions associated with PM_2.5_, PM_10_, and CO exposure, and a lower risk of admissions with NO_2_ exposure, compared to males. The older adults (≥65 years) experienced a higher risk of emergency admissions associated with SO_2_ and O_3_ exposure compared to the non-elderly group. We found significant positive associations between respiratory disease- and cardiovascular disease-related emergency hospital admissions and ambient SO_2_, PM_2.5_, PM_10_, NO_2_, and CO concentrations. Age and gender were identified as effect modifiers of all-cause admissions.

## 1. Introduction

Air pollution poses a major threat to public health [1,2,3]. Previous studies have established a strong exposure-response relationship between short-term increases in air pollution and disease [4,5], especially cardiovascular and respiratory disease [4,5,6]. The World Health Organization estimated that outdoor air pollution caused 4.2 million premature deaths worldwide in 2016 alone [7]. Ambient pollution comprises several measurable pollutants, such as particulate matter (PM) of varying sizes, nitrogen dioxide (NO_2_), sulfur dioxide (SO_2_), carbon monoxide (CO), and ozone (O_3_). There has been increased interest in the extent of the effects on a host of health outcomes, ranging from low birth weight to cancer mortality [8,9].

Examining the impact of air pollution on specific health outcomes is important in discerning mechanistic insights [10,11,12,13,14,15,16,17]. Examining the impact of air pollution, from the perspective of health services utilization, is crucial for demand projection, resource allocation, and quantification of the economic impact. For example, a recent study from the U.K. found that between 2017 and 2025, the total cost of air pollution to the National Healthcare Service was estimated to be £5.56 billion, corresponding to 1.15 million new cases of disease [18]. Studies in various parts of the world have demonstrated a positive association between air quality measures and emergency hospital admissions. A study in Lisbon, Portugal, reported a lagged positive association between CO, NO_2_, and SO_2_, and emergency admissions for cardiorespiratory illness [19]. A study in Taiwan found a positive association between urban air pollution levels (O_3_ and CO) and emergency hospital admission for cerebrovascular disease [20]. In the U.S., PM_2.5_ and O_3_ were found to be positively associated with cardiorespiratory emergency admissions in St. Louis, Missouri [21], and positive associations were observed between pollutants (NO_2_, CO, and black carbon) and myocardial infarction and pneumonia in Boston, Massachusetts [22]. In Beijing, China, two separate studies have reported positive association between ambient air pollution (SO_2_, NO_2_, and PM_10_) and emergency admissions due to cardiovascular and respiratory causes [23,24].

Furthermore, a disproportionate burden of air-pollution-related illnesses is borne by low- and middle-income countries, especially within South-East Asia (SEA) [25]. SEA is populated by some 700 million people—more than the European Union, Latin America, or the Middle East. Despite SEA bearing the largest burden from air pollution, there has been limited understanding of the health impacts in this region [26], as much of the literature has focused on North America and Europe.

This study aimed to investigate the short-term association between ambient air quality and emergency hospital admissions, in order to inform the development and implementation of measures aimed at augmenting hospital resources during anticipated periods of poorer air quality. We hypothesized that exposure to higher pollutant levels was associated with higher daily admissions.

## 2. Materials and Methods

### 2.1. Setting

Singapore is an urbanized island city-state in South-East Asia with a population of 5.9 million, residing within a land area of 728 square kilometers. Singapore has a gross domestic product of S$469.1 billion [27], and a life expectancy at birth of 82.1 years [28]. Singapore is situated close to the equator and has a tropical climate. As a result of its geographic location and maritime exposure, its climate is characterized by uniform temperature and pressure, high humidity, abundant rainfall, and no true distinct seasons. We performed the study using an administrative dataset from Singapore General Hospital (SGH), the largest and oldest tertiary medical center in Singapore, with comprehensive clinical services and over 1700 inpatient beds.

### 2.2. Study Design

In this time-series analysis, we included all emergency admissions at SGH from 2009 to 2017, and examined its short-term association with ambient air pollutant concentrations. We fitted a quasi-Poisson model, and used a distributed lag non-linear model (DLNM) framework, to account for the non-linear lagged effects of the environmental exposures. We also investigated effect modification by sex and age groups.

### 2.3. Exposure Data

In Singapore, ambient pollutant levels are continuously monitored at 22 telemetric air quality monitoring stations across the island [29]. Daily averages of particulate matter of aerodynamic diameter less than 2.5 μm (PM_2.5_) and aerodynamic diameter less than 10 μm (PM_10_), O_3_, SO_2_, NO_2_, CO, and mean daily temperature from 1 January 2009 to 31 December 2017, were obtained from the National Environment Agency (NEA), which is a statutory board under the Ministry of Sustainability and the Environment in Singapore. Public holiday data were obtained from the Ministry of Manpower, Singapore [30]. Ambient temperature data from eleven weather stations located across Singapore were obtained from the Meteorological Service Singapore. We computed the daily arithmetic mean of temperature over all stations. Mid-year population size estimates for each year during the study period were obtained from the Singapore Department of Statistics [31].

### 2.4. Outcome Data

Emergency admissions were defined as inpatient admissions through the emergency department. Annually, the SGH emergency department receives more than 120,000 visits, over 40,000 of which translated to inpatient admissions [32]. All patients above the age of 18 years, admitted via the emergency department at SGH from 1 January 2009 to 31 December 2017 were included in this study. Outcome data were obtained from electronic health records, namely the SingHealth Electronic Health Intelligence System. Data extracted included the daily number of emergency admissions, as well as patient-level characteristics, such as age, sex, and primary ED diagnosis.

### 2.5. Statistical Analysis

All analyses were performed using base R version 4.0.2 (R Foundation, Vienna, Austria). The number of emergency admissions each day could only take on zero or positive values. Hence, we assumed a Poisson distribution in our regression model. As the variance of the outcome measure exceeded that of its mean, we fitted a quasi-Poisson distribution to account for overdispersion. It is known that disease and symptom onset depend not only on exposure to the present day’s air quality exposure, but also on exposure on previous days (i.e., lagged effect) [33]. Not accounting for the lagged exposure effects could result in inaccurate estimates of outcomes. We used a distributed lag non-linear model (DLNM) (dlnm package version 2.4.2) to investigate the short-term immediate and lagged associations between the six recorded individual air pollutants and the number of emergency admissions at SGH on a daily time scale. The DLNM framework allowed us to simultaneously estimate the immediate and lagged non-linear effects of each air pollutant.

Long-term trends, periodic patterns, and holidays may be potential confounders of the relationship between the outcome and the exposures of interest [34]. First, we established a core multivariable regression model by including the daily number of emergency admissions and independent variables to account for the effects of the day of the week (DOW), month, year, long-term trend, statutory public holidays, the day after public holidays, and population offset. We controlled for the long-term trend of daily emergency admissions using linear and quadratic functions of time. Periodic patterns were accounted for by using categorical terms to represent the year, month of the year, DOW, and a binary term to represent holidays. Population changes may influence the number of individuals at risk of emergency admissions. Using annual mid-year population estimates, we fitted a natural cubic spline model in order to obtain interpolated daily estimates of the Singapore population. We then used the base logarithm of the modeled daily estimates as a population offset. Only the independent variables with *p*-values of less than 0.05 were included in the core model.

Using the core model, we individually fitted the cumulative lagged effects of each cross-basis function of the six air pollutants and ambient temperature in seven separate single-pollutant models. We used a natural cubic spline with three degrees of freedom (d.f.) for air pollutant/temperature and a natural cubic spline with three d.f. for lag, to capture the nonlinear air pollutant/temperature effect and the lagged effects from 0 to 7 days. The internal knots of the natural cubic spline were placed by *dlnm* package at the default equally spaced quantiles, while the boundary knots were located at the air pollutant/temperature range.

We then constructed multi-pollutant models to obtain independent adjusted estimates of each pollutant effect. Pearson correlation coefficients were computed on all six air pollutants with each other (Appendix A). Any two air pollutant species with a correlation value of greater than 0.5 were excluded from being in the same model. As a result, three multi-pollutant models were selected with the specified air pollutants: [Model A] PM_2.5_, ozone, NO_2_—estimates for PM_2.5_, ozone, and NO_2_ were taken from this model; [Model B] PM_10_, ozone, NO_2_—estimates for PM_10_ were taken from this model; [Model C] CO, ozone, SO_2_—estimates for CO and SO_2_ were taken from this model. Because NO_2_ and ozone appear in more than one model and their estimated effects are similar from these models (Appendix A), only the above-mentioned estimates are reported in the main text. We then used these separate models to obtain estimates of the daily number of emergency admissions.

In time-series analysis, consecutive observations are more likely to be related than those which are further apart. This phenomenon is known as autocorrelation. If autocorrelation is not accounted for, this may lead to inaccurate estimates of the standard errors of the modeled coefficients. We inspected the Autocorrelation Function and Partial Autocorrelation Function plots from the individual models to assess the degree of autocorrelation. Any residual autocorrelation was accounted for by adding lagged deviance residuals from the full model [35]. The final three models included linear terms consisting of DOW, linear and quadratic functions of time, public holidays, the day after public holidays, and population offset; and non-linear terms consisting of cross-basis functions of PM_2.5_, PM_10_, O_3_, SO_2_, NO_2_, and CO, respectively. Other variables, including temperature, were not included in the final model because they were non-significant. We reported the independent association between each air pollutant and emergency admissions over 7 days using relative risk (RR) with 3 degrees of freedom, and using the median of the daily air pollutant levels as the reference value. The RR represents the change in the risk of emergency admissions, given a change in the exposure referencing its median level. For non-linear associations, the RRs for an air pollutant effect depend on the level of exposure. Therefore, for each air pollutant effect, we selected a moderately higher level of exposure and presented the estimated RR to inform the population health impact during unhealthy air quality events. 

Stratified analyses were performed to evaluate potential effect modification by age, sex, and disease group. Age groups included younger adults, defined as those below 65 years of age, and older adults, defined as those 65 years and above. Disease categories included cardiovascular and respiratory, based on the primary ED diagnosis codes in ICD-9 and ICD-10 (Appendix A). Sensitivity analyses were conducted by varying the degree of flexibility (dfs: 3–5) for the air pollutant cross-basis functions. We assessed statistical significance at the 5% level using Wald tests.

## 3. Results

### 3.1. Descriptive Analysis

There was a mean of 124 emergency admissions per day during the 9-year study period. Patients aged 65 and above constituted 51% of all admissions. There were approximately the same number of male and female patients admitted. The distribution of daily hospital admissions and outdoor air pollutants are shown in Table 1.

### 3.2. Overall Association between Individual Air Pollutants and Emergency Admissions

Figure 1 shows the significant associations (*p*-value < 0.05) between air pollutants and the cumulative risk of total emergency admissions over 7 days. PM_2.5_ and PM_10_ exhibited a U-shaped relationship with admissions risk (Figure 1B,C), while NO_2_ and CO exhibited a complex, non-linear relationship with admissions risk (Figure 1E,F). The relationships between SO_2_, PM_2.5_, PM_10_, NO_2_, CO levels, and emergency admissions were non-linear and positive at higher concentration ranges. Emergency admissions were negatively and non-linearly associated with O_3_ (Figure 1D). For SO_2_ and NO_2_, the cumulative RR for emergency admissions started to increase at higher air pollutant levels (Figure 1A,E). At a SO_2_ concentration of 30 μg/m^3^ (which was below the WHO air quality guideline value), the relative risk of ED admissions was 1.026 (95% CI: 1.023 to 1.029) while at a NO_2_ concentration of 40 μg/m^3^, the relative risk of ED admissions was 1.021 (95% CI: 1.019 to 1.023). Similarly, at a PM_2.5_ concentration of 150 µg/m^3^, the relative risk of ED admission was 1.025 (95% CI: 1.021 to 1.029) (Figure 1B); at a PM_10_ concentration of 200 µg/m^3^, the relative risk of ED admission was 1.036 (95% CI: 1.031 to 1.041) (Figure 1C); at a CO concentration of 3.0 mg/m^3^ (below the WHO air quality guideline value) the relative risk of ED admission was 1.185 (95% CI: 1.170 to 1.199) (Figure 1F). At a higher concentration of O_3_ (50 µg/m^3^), the relative risk of ED admission was 0.978 (95% CI: 0.976 to 0.980), which was a decrease compared to the reference value (Figure 1D). However, the recorded O_3_ concentrations in our data set were all below the WHO air quality guideline value of 100 µg/m^3^. We did not observe a significant relationship between ambient air temperature and emergency admissions. In sensitivity analysis, varying the flexibility of the modeled pollutant-admissions risk relationship did not alter the direction of effect at higher pollutant concentration ranges (Appendix A).

### 3.3. Effect Modification by Sex

For PM_2.5_, PM_10_, NO_2._, and CO, we observed evidence of sex effect modification. Compared to males, females were more susceptible to emergency admissions associated with higher levels of PM_2.5_, PM_10_, and CO (Figure 2B,C,F). Males were more likely to experience emergency admissions associated with higher levels of NO_2_ (Figure 2E). Evidence of sex effect modification for SO_2_ and O_3_ exposure was less clear (Figure 2A,D).

### 3.4. Effect Modification by Age Group

The older adults experienced a higher risk of emergency admissions compared to the younger adults group at higher concentration levels of SO_2_ and O_3_ (Figure 3A,D). Effect modification by age was not evident at higher concentration levels for the other four air pollutants (Figure 3B,C,E,F).

### 3.5. Association between Air Pollutants and Emergency Admission Due to Specific Disease Groups

We observed a statistically significant, positive association between respiratory-related and cardiovascular-related emergency admissions when SO_2_, PM_2.5_, PM_10_, NO_2_, and CO levels tended towards the more extreme end of their maximum value range (Figure 4). At higher concentration levels of PM_2.5_ and PM_10_, the cumulative risk of emergency admissions due to respiratory-related conditions was higher than those due to cardiovascular conditions (Figure 4B,C). We did not observe clear evidence that the risk of NO_2_- or CO-driven admissions differed by respiratory- and cardiovascular-related conditions.

## 4. Discussion

In this study, we examined the associations between air quality variations and the risk of emergency hospital admissions. We also investigated the association between air quality and respiratory- and cardiovascular-related admissions. We found evidence that, at higher concentration levels, five out of the six air pollutants studied were linked to an increased risk of emergency admissions, as well as respiratory and cardiovascular admissions. We identified age group as an effect modifier for the association between SO_2_- and O_3_-driven all-cause emergency admissions; and sex as an effect modifier for the associations between PM_2.5_, PM_10_, NO_2,_, and CO and all-cause emergency admissions. Our study adds a South-East Asian context to the increasing body of evidence on the relationship between air pollution and health, and presents a compelling argument for concerted national efforts and intensified international cooperation to promote and sustain better air quality.

The significant positive associations between SO_2_, PM_2.5_, PM_10_, NO_2_, and CO and cause-specific admissions at higher concentration levels in our study are supported by well-established epidemiological and biological links found in numerous studies [4,5,6,33,36,37]. Our study findings on the positive relationships between measures of air quality and emergency admissions were consistent with those reported in other studies conducted in various parts of the world. A multi-city study conducted in China reported independent, positive associations between air quality (PM_10_, SO_2_, NO_2_, and CO levels) and hospital admissions [38]. A study in Munich, Germany, reported positive relationships between air quality parameters (PM_10_ and NO_2_) and emergency department admissions [39]. A study in Greater Houston, U.S., reported a positive association between PM_2.5_ levels (mean: 12.0 µg/m^3^, lower than that in our study) and emergency department admission [40]. Another study conducted in Chinese cities reported positive relationships between particulate matter concentrations and hospital admissions, though the average pollutant levels far exceeded those recorded in our study (PM_2.5_ of 63 µg/m^3^ vs. 18.7 µg/m^3^, PM_10_ of 99.4 µg/m^3^ vs. 30.2 µg/m^3^). [36]. The U-shaped associations between ED admissions and particulate matter concentrations were unexpected. More studies are required to understand why an initial rise in particulate matter concentrations leads to a decline in ED admission risk, before a rise in risk follows.

Out of the six air pollutants studied, O_3_ was the only pollutant that demonstrated an overall negative association with emergency admissions, regardless of age, sex, or cardiovascular or respiratory conditions as the primary ED diagnosis. Existing studies reporting the relationship between O_3_ and emergency admissions had been inconsistent. For example, a German study found that higher O_3_ concentrations were associated with fewer ED visits in transitional seasons, such as spring and autumn [39]. However, as our study setting, Singapore, is located near the equator and does not exhibit the seasonal variation like the temperate regions, we were not able to verify this relationship. A positive association between O_3_ and admissions, due to cardiac admissions, was found in Hong Kong, but a negative association was found in London [41]. A Canadian study showed that higher O_3_ levels were associated with more asthma ED visits [42]. More studies are required to elucidate the mechanisms of the effect of O_3_ on emergency admissions.

Our study found evidence of significant effect modification by age and sex. Generally, one would have expected the older adults group to be more susceptible to air-pollution-driven health outcomes compared to the younger adults group, principally due to diminishing functional reserve and an increased number of comorbidities in the former. This was our observation for the effect of SO_2_ and O_3_ exposures on the risk of emergency admissions in the present study. Although effect modification by sex in air pollution epidemiology has been widely reported and possible explanations have been explored, the exact sources of the difference remain unclear [43]. Therefore, further studies are needed to understand the mechanisms through which age and sex modifies the effect of pollutant-specific risk of emergency admissions.

While the relationship between air pollutant exposure and adverse health outcomes are clearly established and likely causal, given the large body of epidemiological data, studies are still in progress to increase the knowledge on the intracellular pathways and propose intervention strategies. Several biological mechanisms have been identified to be responsible for air-pollution-dependent cardiovascular and respiratory outcomes. Short-term and long-term exposure to all six air pollutant species in this study can cause direct translocation of air pollutants into circulation, induce sustained pulmonary oxidative stress and systemic inflammation, and increase autonomic system activation [10,11,12,13,14]. These cellular changes can subsequently contribute to increased atherosclerotic burden, increased risk of thrombosis and arrhythmias, and therefore, adverse cardiovascular outcomes [10,15]. In the respiratory system, the same cellular changes can result in increased susceptibility to infection, lung inflammation, and reduced lung function, contributing to the respiratory conditions such as pneumonia, chronic obstructive pulmonary diseases exacerbation, and asthma exacerbation [16,17].

One strength of this study lies in the high-quality data analyzed. We included exposure data that was directly collected by stations that were geographically distributed across Singapore. We used nine successive years of time-series data which allowed the capture of long-term trends and seasonal influences that potentially confound the associations between air quality and health events. There are several limitations to this study. First, the study design did not permit controlling for time-variant variables, such as behavioral changes related to air pollution, because people may take different approaches to mitigation according to their beliefs and attitudes. The community uptake of recommended protective measures during periods of regional haze may have altered the risk of admissions. Second, in taking the event date to be the date of presentation to the emergency department, there was potential for misclassification of the exposure with the onset of symptoms. To account for this, we computed the cumulative risk of admissions associated with immediate and lagged pollutant exposures from 0 to 7 days. Third, not all ED admissions may have been causally related to air pollution exposures, though our findings on the positive association between air quality and respiratory-specific and cardiovascular-specific related admissions have been well established in several settings across the globe. Assessing the overall impact of air quality variations on the volume of admissions has implications for health system management. Finally, we only used data from a single center, and therefore, they may not be generalizable at the national level.

Our study demonstrates that even when average CO levels are within the desired levels prescribed by the WHO (4 mg/m^3^), a daily increase in CO concentrations can nonetheless be associated with a corresponding increase in the risk of all-cause emergency admissions. The ubiquitous and involuntary exposure of the entire population to prevalent air quality can greatly magnify the health burden. Understanding the relationship between air quality and emergency admissions is a crucial step in quantifying the actual healthcare burden, and therefore, the negative externality which air pollution causes to the regional economy. Our study findings can inform the development and timing of measures aimed at improving resource allocation to emergency healthcare systems during anticipated periods of poorer air quality. They may also be useful in targeting mitigation measures for vulnerable population sub-groups.

## 5. Conclusions

We found evidence of positive associations between five individual air pollutants at higher concentration levels and respiratory-related and cardiovascular-related emergency hospital admissions from a major hospital in Singapore. We also identified age and sex as effect modifiers of all-cause admissions for several air pollutants. Our study provides additional insight upon which further research into the air quality effects on ED admissions can be conducted. More studies are required to examine the relationship between air quality and emergency hospital admissions at the national level.

## Figures and Tables

**Figure 1 ijerph-19-13336-f001:**
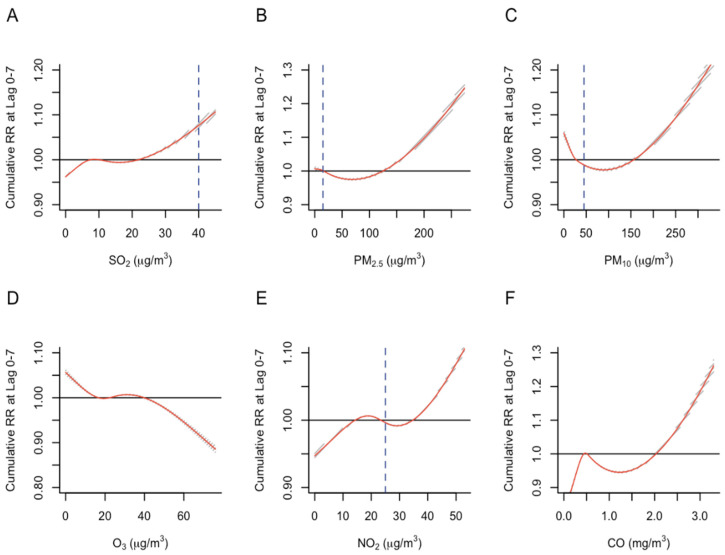
Association between air pollutant levels and cumulative relative risk (RR) of emergency admissions. (**A**) SO_2_ levels and RR; (**B**) PM_2.5_ levels and RR; (**C**) PM_10_ levels and RR; (**D**) O_3_ levels and RR; (**E**) NO_2_ levels and RR; (**F**) CO levels and RR. The solid red line represents the modeled cumulative risk. The shaded areas represent the 95% confidence intervals. The horizontal solid black line represents the reference value. The vertical blue dashed line indicates the air pollutant level recommended by WHO air quality guidelines. There are no vertical dashed lines for Figure 1D,F because the maximum values observed over the study period were much lower than the WHO guideline values for ozone (100 µg/m^3^) and carbon monoxide (4 mg/m^3^).

**Figure 2 ijerph-19-13336-f002:**
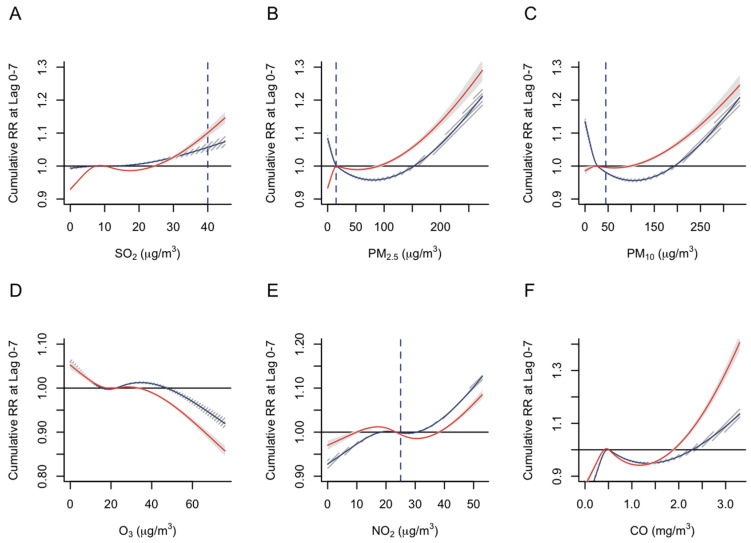
Association between air pollutant levels and cumulative relative risk (RR) of emergency admissions stratified by sex. (**A**) SO_2_ levels and RR; (**B**) PM_2.5_ levels and RR; (**C**) PM_10_ levels and RR; (**D**) O_3_ levels and RR; (**E**) NO_2_ levels and RR; (**F**) CO levels and RR. The solid blue lines represent the modeled cumulative risk for males, while the solid red lines represent the modeled cumulative risk for females. The shaded areas represent the 95% confidence intervals. The horizontal solid black line represents the null value. The vertical blue dashed line represents the desired air pollutant levels indicated in the WHO air quality guidelines.

**Figure 3 ijerph-19-13336-f003:**
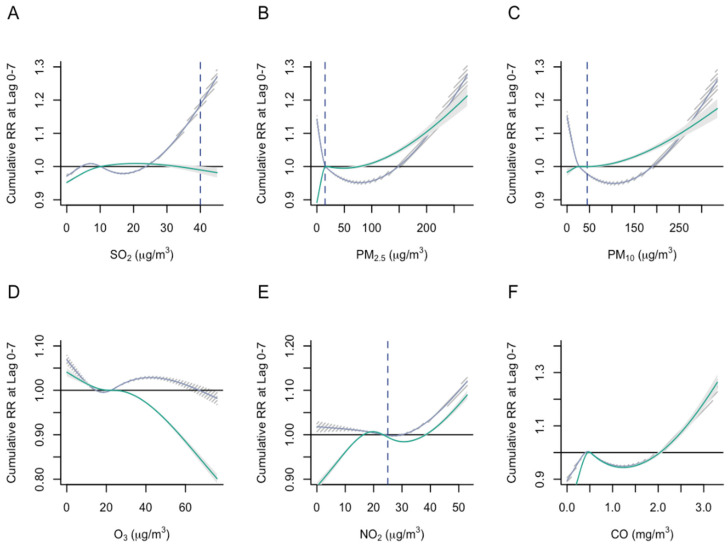
Association between air pollutant levels and cumulative relative risk (RR) of emergency admissions stratified by age group. (**A**) SO_2_ levels and RR; (**B**) PM_2.5_ levels and RR; (**C**) PM_10_ levels and RR; (**D**) O_3_ levels and RR; (**E**) NO_2_ levels and RR; (**F**) CO levels and RR. The solid gray line represents the modeled cumulative risk for the older adults, while the solid green line represents the modeled cumulative risk for the younger adults. The shaded areas represent the 95% confidence intervals. The horizontal solid black line represents the reference value. The vertical blue dashed line represents the desired air pollutant levels indicated in the WHO air quality guidelines.

**Figure 4 ijerph-19-13336-f004:**
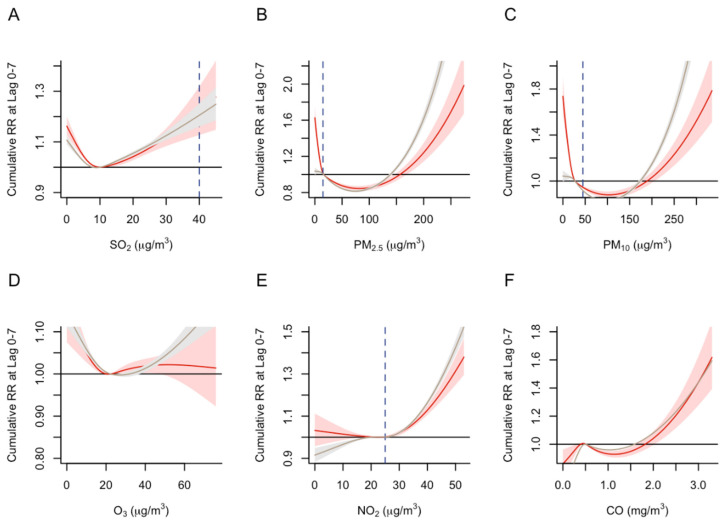
Association between air pollutant levels and cumulative relative risk (RR) of emergency admissions due to cardiovascular and respiratory conditions. (**A**) SO_2_ levels and RR; (**B**) PM_2.5_ levels and RR; (**C**) PM_10_ levels and RR; (**D**) O_3_ levels and RR; (**E**) NO_2_ levels and RR; (**F**) CO levels and RR. The solid red line represents the modeled cumulative risk for cardiovascular-related conditions, while the solid gray line represents the modeled cumulative risk for respiratory-related conditions. The shaded areas represent the 95% confidence intervals. The horizontal solid black line represents the null value. The vertical blue dashed line represents the desired air pollutant levels indicated in the WHO air quality guidelines.

**Table 1 ijerph-19-13336-t001:** Summary of daily measures of emergency hospital admissions and air pollutants, from 2009 to 2017.

	Daily Mean (SD)	Quartiles				
		Min	Q1	Median	Q3	Max
Number of daily admissions						
Total	124.4 (18.2)	69	112	124	136	195
Number of admissions by age						
Age < 65	63.5 (11.1)	32	56	63	71	106
Age ≥ 65	60.9 (11.4)	26	53	60	68	106
Number of admissions by gender						
Male	62.1 (10.6)	31	55	62	69	107
Female	62.3 (10.7)	29	55	62	69.5	101
Air pollutants, mean daily						
PM_2.5_ (µg/m^3^)	18.7 (13.5)	5.1	12.8	16.1	20.8	274.4
PM_10_ (µg/m^3^)	30.2 (16.5)	9.7	22.8	27.3	33.6	335.9
O_3_ (µg/m^3^)	24.1 (9.9)	4.4	17.1	22.3	29.5	76.0
NO_2_ (µg/m^3^)	23.9 (7.1)	6.8	19.0	23.4	28.5	53.4
SO_2_ (µg/m^3^)	10.8 (6.2)	2.0	5.6	10.2	14.1	45.9
CO (mg/m^3^)	0.5 (0.2)	0.2	0.4	0.5	0.6	3.3
Meteorological factors						
Mean daily temperature (°C)	27.8 (1.1)	23.4	27.1	27.9	28.7	30.8
Admissions by diagnosis						
Respiratory	16.6 (5.1)	3	13	16	20	44
Cardiovascular	6.7 (3.0)	1	5	6	8	18

## Data Availability

The data that support the findings of this study are available from Singapore Health Services and National Environmental Agency. But restrictions apply to the availability of these data, which were used under license for the current study, and so are not publicly available. Data are however available from the authors upon reasonable request and with permission of Singapore Health Services and National Environmental Agency.

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
