# Peer review of "Ambient Air Quality and Emergency Hospital Admissions in Singapore: A Time-Series Analysis"

_ijerph, 2022, doi:10.3390/ijerph192013336_

Round 1

Reviewer 1 Report

This paper investigate the correlation between air quality and emergency hospital admissions in Singapore using statistical analysis method. The methods are well designed and adequately described. The results are clearly presented and discussed. However, the Introduction section is a little insufficient. Overall, the topic of this paper is highly relevant to Environmental Research and Public Health. Therefore, the publication of this paper in IJERPH is recommended. Please see detailed comments below.

1. More background is suggested. For example, page 2 second paragraph, references about recognizing the mechanistic insights is recommended. Also, the study by Sudarshan et al seems irrelevant to current topic. 

2. The relationships between measures of air quality and emergency admissions in other places of the world can be included in the Introduction. 

Author Response

Point 1: More background is suggested. For example, page 2 second paragraph, references about recognizing the mechanistic insights is recommended. Also, the study by Sudarshan et al seems irrelevant to current topic.

Point 2: The relationships between measures of air quality and emergency admissions in other places of the world can be included in the Introduction.

Response to Points 1 & 2: References on recognizing the mechanistic insights have been added. The reference to study by Sudarshan et al was removed (see strikethrough below). A few examples of the relationship between measures of air quality and emergency admissions in other places of the world were included in the introduction section, with references to studies done in China, USA, and Europe.

“Examining the impact of air pollution on specific health outcomes is important in discerning mechanistic insights.[10–17] Examining the impact of air pollution from the perspective of health services utilization is crucial for demand projection, resource allocation and quantification of the economic impact. For example, a recent study from the UK found that between 2017-2025, the total cost of air pollution to the National Healthcare Service was estimated to be £5.56 billion, corresponding to 1.15 million new cases of disease.[18] Studies in various parts of the world have demonstrated a positive association between air quality measures and emergency hospital  admissions. A study in Lisbon, Portugal, reported a lagged positive association between CO, NO2, and SO2, and emergency admissions for cardiorespiratory illness.[19] A study in Taiwan found a positive association between urban air pollution levels (O3 and CO) and emergency hospital admission for cerebrovascular disease.[20] In the USA, PM2.5 and O3 were found to be positively associated with cardiorespiratory emergency admissions in St. Louis, Missouri;[21] positive associations were observed between pollutants (NO2, CO, and black carbon) and myocardial infarction and pneumonia in Boston, Massachusetts.[22] In Beijing, China, two separate studies have reported positive association between ambient air pollution (SO2,NO2 and PM10) and emergency admissions due to cardiovascular and respiratory causes.[23,24] In another study, Sudarshan et al utilized environmental data to develop forecasting models of Emergency Department (ED) visits, which enables dynamic deployment of scarce healthcare resources (e.g. manpower rostering, supplies and logistics) and obtained prediction errors ranging from 8-10%.[11]

Reviewer 2 Report

The manuscript presents a study on a very important and current public health problem. The material is extensive and covers a significant period of time and database. I would recommend that the authors re-analyze the literature on this issue to include all available data. 

1. What is the main question addressed by the research?

The manuscript presents a research that aim to investigate the association between air quality and emergency hospital admissions in Singapore. 

2. Do you consider the topic original or relevant in the field, and if so, why?

I think the topic is relevant because environmental pollution leads to long-term health problems and the presence of more evidence of this could lead to more serious public measures being taken to limit air pollution with harmful emissions.

3. What does it add to the subject area compared with other published material?

The study included a specific geographic region with significant urbanization and a large population. Significant correlations of certain pollutants with increased hospitalizations were found in different age groups and gender.

4. What specific improvements could the authors consider regarding the methodology?

More extensive analyzes can be done for individual specific cardiovascular and pulmonary diseases

5. Are the conclusions consistent with the evidence and arguments presented and do they address the main question posed?

Yes, the conclusions correspond to the set aim of the study and the analyzes made

6. Are the references appropriate?

The references are appropriate

7. Please include any additional comments on the tables and figures.

Figures could be colored for better visualization

Author Response

Point 1:  More extensive analyzes can be done for individual specific cardiovascular and pulmonary diseases.

Response:  In our study, daily counts for cardiovascular admissions were 16 per day while those for respiratory-related admissions were about 6 per day. Specific cardiovascular and pulmonary disease analyses would substantially reduce the statistical power given the single-centre data we used for this study. We agree with the reviewer on the utility of stratified analyses and will explore this when we acquire national level data.

Point 2:  Figures could be colored for better visualization.

Response: For Figure 4, we have adjusted the colors of overlapping confidence intervals to give better contrast.

Reviewer 3 Report

According to the author of the manuscript air pollution exposure may raise emergency healthcare demand, especially in South-East Asia, where air pollution-related health consequences are considerable. This article examines the link between air quality and Singapore hospital admissions. Quasi-Poisson regression was used with a distributed lag non-linear model (DLNM) to investigate short-term relationships between air quality fluctuations and emergency hospital admissions in Singapore during 2009-2017. Higher SO2, PM2.5, PM10, NO2, CO concentrations increased the likelihood of all-cause, cardiovascular, and respiratory emergency admissions over 7 days. Increased O3 levels led to a non-linear decline in emergency admissions. Females had a greater risk of emergency admissions from PM2.5, PM10, and CO exposure than men, but a lower risk from NO2. Older people (65) had a greater incidence of SO2 and O3-related emergency admissions than younger adults. The manuscript identified favorable correlations between respiratory and cardiovascular hospital admissions and SO2, PM2.5, PM10, NO2 and CO. According to the empirical evidences age and gender affect all-cause hospitalizations.

The novelty of the manuscript and contribution to the literature is low.

Please consider the following

The authors should provide the theoretical background of the methodology used.

The estimated parameters with significance levels and hypothesis tests should be provided.

Please pay more attention to the statistical analysis and theory of the manıscript.

Author Response

Point 1:  The authors should provide the theoretical background of the methodology used.

Response: We have added the following points within the “statistical analysis” section to explain the theoretical background of the methodology used.

  1. Why we chose a quasi-Poisson distribution model:

The number of emergency admissions each day could only take on zero or positive values. Hence we assumed a Poisson distribution in our regression model. As the variance of the outcome measure exceeded that of its mean, we fitted a quasi-Poisson distribution to account for overdispersion in the outcome measure.

  1. Why we chose DLNM to account for the lagged exposure effects:

Not accounting for the lagged exposure effects could result in inaccurate estimates on their effects. We used a distributed lag non-linear model (DLNM) (dlnm package version 2.4.2) to investigate the short-term immediate and lagged associations between the six recorded individual air pollutants and the number of emergency admissions at SGH on a daily time scale.

  1. Justifications for adjusting for the confounding variables to account for periodic patterns:

Periodic patterns were accounted for by using categorical terms to represent the year, month of the year, DOW and a binary term to represent holidays. Population changes may influence the number of individuals at risk of emergency admissions.

  1. Justification for adjusting for autocorrelation:

In time-series analysis, consecutive observations are more likely to be related that those which are further apart. This phenomenon is known as autocorrelation. If autocorrelation is not accounted for, this may lead to inaccurate estimates of the standard errors of the modelled coefficients.

Point 2:  The estimated parameters with significance levels and hypothesis tests should be provided.

Response: In our study, we computed the relative risk (RR) to reflect the strength of association between ED admissions and the air pollutants. As the modelled relationships were non-linear, meaning that the RR changes according to the exposure level, it would not have been practical to produce a table with the RRs and their corresponding 95% confidence intervals for every exposure level. Instead, we produced figures depicting the dose-response relationship, which is a common practice in many published studies that use a similar approach towards non-linear estimations of health and air quality relationships. However, we also recognize that some quantitative estimates may be beneficial. Therefore, for each air pollutant effect, we selected a moderately higher level of exposure and presented the estimated RR to inform the population health impact during unhealthy air quality events.

We added the following line at the end of the Methods section to inform readers of the level of significance we considered, and the statistical tests used:

“We assessed statistical significance at the 5% level using Wald tests.”

We also indicated more clearly in the text in sections 2.2 and 2.5 where results were found to be statistically significant.

Point 3:  Please pay more attention to the statistical analysis and theory of the manuscript.

Response 3: Similar to point 1 of Reviewer 3, we have added as many points of explanation as possible to provide sufficient theoretical background of the statistical method.

Round 2

Reviewer 3 Report

The manuscript can be published in this form